# Low Light Increases the Abundance of Light Reaction Proteins: Proteomics Analysis of Maize (*Zea mays* L.) Grown at High Planting Density

**DOI:** 10.3390/ijms23063015

**Published:** 2022-03-10

**Authors:** Bin Zheng, Wei Zhao, Tinghu Ren, Xinghui Zhang, Tangyuan Ning, Peng Liu, Geng Li

**Affiliations:** College of Agronomy, Shandong Agricultural University, Tai’an 271018, China; zhengbin218td@sina.com (B.Z.); zhaoxwei9298@sina.com (W.Z.); rthu123@sina.com (T.R.); zhangxinghui98@sina.com (X.Z.); ningty@sdau.edu.cn (T.N.); liup@sdau.edu.cn (P.L.)

**Keywords:** photosynthesis, low light, chlorophyll fluorescence, proteomics, maize, dense planting

## Abstract

Maize (*Zea mays* L.) is usually planted at high density, so most of its leaves grow in low light. Certain morphological and physiological traits improve leaf photosynthetic capacity under low light, but how light absorption, transmission, and transport respond at the proteomic level remains unclear. Here, we used tandem mass tag (TMT) quantitative proteomics to investigate maize photosynthesis-related proteins under low light due to dense planting, finding increased levels of proteins related to photosystem II (PSII), PSI, and cytochrome *b*_6_*f*. These increases likely promote intersystem electron transport and increased PSI end electron acceptor abundance. OJIP transient curves revealed increases in some fluorescence parameters under low light: quantum yield for electron transport (φE_o_), probability that an electron moves beyond the primary acceptor Q_A_^−^ (ψ_o_), efficiency/probability of electron transfer from intersystem electron carriers to reduction end electron acceptors at the PSI acceptor side (δR_o_), quantum yield for reduction of end electron acceptors at the PSI acceptor side (φR_o_), and overall performance up to the PSI end electron acceptors (PI_total_). Thus, densely planted maize shows elevated light utilization through increased electron transport efficiency, which promotes coordination between PSII and PSI, as reflected by higher apparent quantum efficiency (AQE), lower light compensation point (LCP), and lower dark respiration rate (R_d_).

## 1. Introduction

To improve yield per acre, farmers often plant modern crops at high densities [1]. In plants growing at high density, the upper leaves and neighboring plants shade the lower leaves, resulting in a low-light environment. Low-light conditions result in morphological and physiological changes in plants, including thinner leaves, lower leaf chlorophyll contents, lower nitrogen contents, lower photosynthetic system activities, and lower enzyme activities [2,3,4,5,6]. These changes severely restrict leaf photosynthetic capacity and thus crop production. In maize (*Zea mays* L.), chlorophyll content and photosynthetic capacity decrease significantly as plant density increases, mainly due to changes in chloroplast morphology and damage to leaf ultrastructure, such as fewer and smaller chloroplasts, fewer grana lamellae, and increased damage to mesophyll cell membranes [7,8]. Moreover, for maize grown in the North China Plain (32~40° N; 114~121° E) or other locations at a similar latitude, hours of sunshine and solar radiation decline during the later stages of growth, severely restricting maize production [9,10,11]. Therefore, low light is an unavoidable problem for maize production in many regions, including the North China Plain.

The effects of low-light stress on plant growth and development have been extensively studied. Generally, different crops have different photosynthetic responses to low-light stress. For example, the lower photosynthetic capacity of cucumber (*Cucumis sativus*) leaves under low-light stress is probably due to down-regulated Rubisco gene expression at the transcript and protein levels and decreased initial and total activity as well as activation state of Rubisco [12]. In rice (*Oryza sativa*), low light markedly down-regulates the abundance of chloroplast proteins, especially those involved in carbon fixation (the Calvin–Benson cycle), electron transport, and the ATPase complex [13]. Similarly, proteins related to the Calvin–Benson cycle are significantly down-regulated in maize under low-light stress [14]. Although different species have different responses to low-light stress, the key response mechanisms focus on the important pathway from light energy capture to carbohydrate formation.

Plants adjust their growth to their light environment through means ranging from whole-plant morphological changes to alterations in the stoichiometry of the photosynthetic apparatus [15,16]. As leaves are the main organ of light interception and carbon assimilation, leaf performance greatly affects photosynthetic production. Levels of photosystem II (PSII), the cytochrome *b*_6_*f* complex, ATP synthase, and Calvin–Benson cycle components (especially Rubisco) decrease substantially under low-light stress, severely restricting leaf photosynthetic capacity and thus crop production [5,6,17]. Photosynthetic yield under low-light stress depends on the efficiency of light energy capture by antenna pigments and the delivery of this energy to the photosystem reaction centers [18], which can (to some extent) be adjusted by the plant. For example, shaded ginger (*Zingiber officinale*) leaves maintain photosynthetic production by increasing the efficiency of photosynthetic electron utilization through increased abundance of proteins related to the light-harvesting complex, the oxygen evolution complex, plastocyanin, and ferredoxin-NADP+ reductase (FNR) [19]. Moreover, low-light conditions increase the light-capture efficiency of PSII in soybean (*Glycine max* L. Merr.) leaves but decrease the capacity to transmit energy from PSII to PSI, resulting in decreased photosynthetic capacity [20]. Excess light energy from PSII is channeled by a number of different processes including photochemistry, heat dissipation, and chlorophyll fluorescence [21]. Chlorophyll fluorescence is an important parameter that changes under different growth environments and photosynthetic conditions [21,22]. The JIP test, a chlorophyll fluorescence analysis [23], is widely used to test plant performance under stressful conditions [24,25,26,27], including chilling [28,29], salinity [30], metal [31], and drought stress [30,32].

Maize is an important crop worldwide for food, fodder, and bioenergy. Globally, continuing population increases and consequent economic growth are expected to gradually increase the demand for maize [33,34]. Given relatively stable arable land area and fertilizer application levels, ensuring food security will require enhancing maize production by various methods, including by increasing planting density though the development of density-resistant varieties [35,36,37,38]. Previous studies showed that the root morphology of many crops changes with increased planting density, with increased vertical extension of the root system and decreased horizontal extension [39,40,41]. Generally, root weight and root surface area per plant also decrease in response to dense planting, especially the significant decreases in the activity of the upper root system, which affect nutrient and water uptake and utilization, thus intensifying competition among plants [40,42,43,44]. In addition, high planting density most likely causes a closed field canopy and poor ventilation, decreasing the gas exchange rate and increasing the temperature within the canopy [45,46]. Under dense planting conditions, these changes affect the normal physiological function of individual plants and are reflected in changes in leaf morphology, such as decreased leaf area, thickness, and weight [8,40,47]. Physiologically, chlorophyll content decreases significantly with increased planting density, with fewer number and volume of thylakoid grana lamellae in chloroplasts, ultimately decreasing the leaf photosynthetic performance [7,8]. A theoretical analysis of crop performance under low light suggested that at high planting densities, the effects of light intensity were more significant than that of individual variation on the various physiological functions of crop leaves, especially the apparent photosynthesis rate [48]. Leaves in light-limited conditions are estimated to contribute about 50% of total grain yield; thus, studying how light-limited leaves use light provides critical information for improving crop yield potential [49,50]. Although previous researchers have examined differences in the photosynthetic performance of maize in the North China Plain under shading or dense planting conditions at the physiological and proteomics levels, these studies mostly regarded PSI and PSII as a whole, ignoring the effects of different photosystems on photochemical reactions and photosynthetic performance [14,51]. Therefore, in this study, we used an integrated approach, including physiological analysis and a tandem mass tag (TMT) quantitative proteomics analysis, to explore changes in the abundance of proteins related to photosynthesis under low-light stress induced by high planting density, especially proteins related to light reactions.

## 2. Results

### 2.1. Leaf Physiological and Photosynthetic Parameters

A significant decrease in the leaf mass per area (LMA) and leaf total chlorophyll content (Chl (*a* + *b*)) occurred with increasing planting density in each growth stage (Figure 1A,B). Compared to plants grown at low density (LD), the plants in the normal density (ND) and high density (HD) groups showed significantly lower LMA, by an average of 22.0% to 30.8% over the five growth stages, as well as significantly lower Chl (*a* + *b*), by an average of 7.7% to 15.3% over the five growth stages. The leaf N content in ND and HD plants was significantly lower than that in LD plants at 0, 10, and 20 d after anthesis (Figure 1C). However, at 30 d after anthesis, the leaf N content in ND plants was significantly higher than that in LD leaves, and at 40 d after anthesis, the leaf N content in ND and HD plants was significantly higher, namely 14.1% and 4.1%, respectively, than that in LD leaves.

Under high-light (HL, 1600 μmol m^−2^ s^−1^) conditions, the value of P_n_ was significantly lower in ND and HD leaves than that in LD leaves at 0 d; however, these differences among three planting densities decreased or even reversed as the plants senesced (Figure 2A), with the changes of P_n_ under low-light (LL, 300 μmol m^−2^ s^−1^) conditions displaying a similar trend (Figure 2D). The trend for G_s_ with planting density was similar to that of P_n_, but C_i_ showed an opposite trend to P_n_ and G_s_ (Figure 2B,C,E,F). The higher P_n_ was always accompanied by higher G_s_ but a lower C_i_; hence, the variation in leaf P_n_ among three planting densities was caused by non-stomatal factors.

Results from the light response curves, expressed using a non-rectangular hyperbola model, are shown in Table 1. The maximum gross photosynthetic rates (A_max_) of leaves from ND and HD plants were only significantly higher than that of LD leaves at 30 d (23.3%, 33.2%) and 40 d (29.1%, 45.8%) after anthesis. In contrast, the rate of dark respiration (R_d_) decreased significantly in the order LD > ND > HD within each growth stage (Table 1). The apparent quantum efficiency (AQE), representing the utilization efficiency of low light, increased in the order LD < ND < HD, with significant differences at 20, 30, and 40 d after anthesis. In addition, the light compensation point (LCP) decreased in the order LD > ND > HD at 10, 20, 30, and 40 d after anthesis; however, the maximum LCP was observed in ND at 0 d after anthesis. A similar pattern was observed for the light saturation point (LSP).

### 2.2. Chlorophyll Fluorescence

The chlorophyll fluorescence (ChlF) curve for all maize leaves followed typical OJIP transients when plotted on a logarithmic time scale (Figure 3A–C). Although the overall shapes of the ChlF curves were similar within each treatment for the different growth stages, the induction steps J and I were different. The differential curve (ΔV_t_) was used to visualize and analyze the differences throughout the induction time (Figure 3A–C). We observed positive ΔV_t_ values, indicating a lower efficiency of electron transport, at all four time-points (10, 20, 30, and 40 d after anthesis) for all three densities; moreover, compared to LD, the ΔV_t_ values in ND and HD were significantly lower within each growth stage.

The phenomenological energy fluxes per excited cross section (CS) of the maize leaves after anthesis are shown in Figure 4. At day 0, the absorbed energy (ABS/CS), trapped energy (TR/CS), electron transport (ET/CS), dissipated energy (DI/CS), and active reaction centers (RCs) per cross section (RC/CS) were significantly higher in ND and HD than in LD plants. However, from 10 to 40 d after anthesis, the values of each parameter decreased in the order ND > HD > LD (Figure 4). The most visible changes over the whole experiment were found for RC/CS and ET/CS.

For most biophysical parameters, including the quantum yield variables, specific activities per RC and performance indexes (PI), we normalized the ND and HD results to those of LD leaves (Figure 5). Similar φP_o_ values were observed among the different treatments within each growth stage; that is, the maximum yield of the primary photochemistry of PSII was not affected by plant density. However, there were differences in W_k_ (ratio of variable fluorescence F_k_ to the amplitude F_j_ − F_0_), V_j_ (relative variable fluorescence at the J-step), ψ_o_ (probability that an electron moves beyond the primary acceptor Q_A_^−^), φE_o_ (quantum yield for electron transport Q_A_^−^ to plastoquinone), and φR_o_ (quantum yield for reduction of end electron acceptors at the PSI acceptor side). The W_k_ and V_j_ values in ND and HD were significantly lower than those in LD, except for the W_k_ values in HD at 0 and 10 d (Figure 5A,B), indicating that the performances of the PSII donor and acceptor side were improved by dense planting. This conclusion is supported by the results for ψ_o_. The ψ_o_ value increased in the order LD < HD < ND within each growth stage (Figure 5), indicating that the improvement on the donor side was higher than that on the acceptor side. A similar pattern was observed for φE_o_ and φR_o_. We found that the efficiency/probability with which an electron from the intersystem electron carriers was transferred to RE (δR_o_) also increased in the order LD < HD < ND within each growth stage (Figure 5), suggesting that the electron transport chain between PSII and PSI was improved by dense planting. At 0 and 40 d, the φD_o_ (quantum yield of energy dissipation) value increased in the order ND < LD < HD, although the differences were not significant (Figure 5A,E); at 20 and 30 d, however, it increased in the order HD < ND < LD, with significant differences between HD and LD (Figure 5C,D). Moreover, φD_o_ values were similar among the three densities at 10 d (Figure 5B). We inferred from these φD_o_ values that the quantum yield of energy dissipation was affected by both plant density and growth stage.

In addition, we found that planting density affected the ratio between the energy absorption by the antenna light-harvesting complex (ABS) and by the active PSII RC, decreasing in the order LD > HD > ND; these differences were significant, except at 0 d after anthesis (Figure 5). The decrease in ABS/RC was accompanied by a decrease in trapping per active PSII RC (TR_o_/RC). The maximum electron transport flux per active PSII RC (ET_o_/RC) was observed in HD leaves at each growth stage, while the maximum electron transport from Q_A_^−^ to the PSI electron acceptors (RE_o_/RC) was observed in HD leaves at 0 and 10 d after anthesis but in ND leaves at 20–40 d after anthesis (Figure 5). Moreover, the energy dissipation (DI_o_/RC) of HD was significantly lower than that of LD leaves in each growth stage, except at 0 d after anthesis.

PI_abs_ is often used to quantify the overall performance of PSII, while PI_total_ reflects the performance of the PSII electron donor side up to RE, which is involved in several electron transport processes, such as those represented by φP_o_, ET_o_/TR_o_ and RE_o_/ET_o_. In the present study, the PI_abs_ and PI_total_ decreased in the order ND > HD > LD within each stage. Moreover, the ΔI/I_o_, which is a measure of the maximum redox activity of the PSI RC, showed a similar pattern to PI_abs_. Finally, the coordination between PSII and PSI (Φ_(PSI/PSII)_) in ND and HD plants was higher than that in LD plants (Figure 5).

### 2.3. TMT-Based Quantitative Proteomics Analysis in Maize Leaves

We identified 10,031 proteins in the maize leaves by TMT-based quantitative proteomics analysis and obtained relative quantitative information for 9226 proteins (Appendix A). Proteins with a *p*-value < 0.05 (Student’s *t*-test) and a fold-change ratio >1.30 or <0.77 were considered differentially abundant proteins (DAPs). At 0 d, 77 DAPs were identified in ND vs. LD, including 29 up- and 48 down-regulated proteins, of which 30 proteins were specifically enriched in this group (Figure 6A,B). In HD vs. LD, 248 DAPs were differentially enriched, containing 138 up- and 110 down-regulated proteins, of which 135 proteins were specifically enriched in this group (Figure 6A,B). Furthermore, 33 DAPs were commonly shared in both ND and HD (Figure 6A), of which 13 DAPs were up-regulated, 19 DAPs were down-regulated, and 1 DAP showed the opposite abundance trend (Figure 6C). At 20 d, 217 DAPs were identified in ND vs. LD, including 57 up- and 160 down-regulated proteins, of which 138 proteins were specifically enriched in this group (Figure 6A,B). In HD vs. LD, 247 DAPs were differentially enriched, containing 172 up- and 75 down-regulated proteins, of which 130 proteins were specifically enriched in this groups (Figure 6A,B). Furthermore, 38 DAPs were commonly shared in both ND and HD (Figure 6A), of which 9 DAPs were up-regulated, 28 DAPs were down-regulated, and 1 DAP showed the opposite abundance trend (Figure 6C). At 40 d after anthesis, 94 DAPs were identified in ND vs. LD, including 40 up- and 54 down-regulated proteins, of which 31 proteins were specifically enriched in this groups (Figure 6A,B). In HD vs. LD, 295 DAPs were differentially enriched, containing 231 up- and 64 down-regulated proteins, of which 77 proteins were specifically enriched in this groups (Figure 6A,B). Furthermore, 25 DAPs were commonly shared in both ND and HD (Figure 6A), of which 11 DAPs were up-regulated, 7 DAPs were down-regulated, and 7 DAPs showed the opposite abundance trend (Figure 6C).

Subcellular localization of all the identified DAPs showed that 4003 proteins (40.1%) were located in the chloroplasts, 2425 (24.3%) in the cytoplasm, 1986 (19.9%) in the nucleus, 521 (5.2%) in the mitochondria, 495 (5.0%) in the plasma membrane, 177 (1.8%) extracellular, 129 (1.3%) in the vacuolar membrane, 103 (1.0%) in the cytoskeleton, 73 (0.7%) in the endoplasmic reticulum, 36 (0.4%) in the peroxisome, and 31 (0.3%) in the Golgi apparatus (Figure 6D). Thus, chloroplast proteins, which include many photosynthetic proteins, were most affected by planting density.

### 2.4. Gene Ontology (GO) Classification of DAPs

To further understand the nature of the identified and quantified DAPs (fold-change ratio >1.30 or <0.77), we annotated their functions and features using GO enrichment analysis. The DAPs were grouped into three hierarchically structured GO terms: biological process (Figure 7A), cellular component (Figure 7B), and molecular function (Figure 7C). There were significant differences in GO terms at different time points.

At 0 d, GO cluster analysis showed that the DAPs in ND vs. LD were highly enriched in macromolecular complex assembly, and transmembrane transport. These DAPs were located in ATPase and peptide complex and play roles in acyltransferase activity. The DAPs in HD vs. LD were highly enriched in ion homeostasis, response to light intensity, response to hydrogen peroxide, and nuclear transport. These DAPs were mainly located in the ribonucleoprotein complex, membrane-bound organelles, and the PSII RC and were involved in ferric ion activity, DNA transcription and regulation, and polygalacturonase activity. Nine significantly enriched GO terms were found among the DAPs of HD vs. ND, which were located in the macromolecular complex, e.g., ribosomes, proteasomes, and nucleus, and play roles in the structural constituents of ribosome and molecule activity, nucleic acid binding, and acid phosphatase activity (Figure 7).

At 20 d, the DAPs in ND vs. LD were highly enriched in metabolic and biosynthetic process, as well as photosynthesis and light harvesting. These DAPs were mainly located in the thylakoids, chloroplasts, photosystems, and mitochondria and function in chlorophyll binding, electron carrier activity, and copper ion binding. The DAPs in HD vs. LD were highly enriched in lignin metabolic and biosynthetic process; phenylpropanoid biosynthetic process; regulation of post-embryonic development; organic acid catabolic process; and peptide, amide, and metal ion transport. They were mainly located in the photosystems, glycine cleavage complex, and plasma membrane and were involved in transporter, ATPase, hydrolase, oxidoreductase, NAD(P)H dehydrogenase (quinone), and transferase activity, as well as FMN binding. However, GO analysis showed that many more biological process categories and molecular function categories were highly abundant in the DAPs in HD vs. ND. Specifically, the DAPs in HD vs. ND were highly enriched in peptide, amide, and metal transport, as well as positive regulation of macromolecule metabolic and biosynthetic processes, and play roles in transporter activity and electron carrier activity (Figure 7).

At 40 d, the DAPs in ND vs. LD were highly enriched in the glycerolipid metabolic process, which were mainly located in mitochondria and respiratory chains, and have roles in transition metal ion binding and nucleic acid binding. The DAPs in HD vs. LD were highly enriched in transmembrane transport, cellular carbohydrate catabolic process, and proteolysis. These DAPs were mainly located in the cell wall, external encapsulating structures, and the ATPase synthase complex and are involved in enzyme and transporter activities. Lastly, the DAPs in HD vs. ND were highly enriched in the compound catabolic process and response to anoxia. They were mainly located in the cell wall, external encapsulating structures, membrane, and extracellular region and play roles in compound binding, ion binding, and enzyme activities (Figure 7).

### 2.5. Kyoto Encyclopedia of Genes and Genomes (KEGG) Pathway Analysis of DAPs

We used the KEGG database to identify enriched pathways using a two-tailed Fisher’s exact test to determine the enrichment of DAPs against all identified proteins (*p*-value < 0.05). These pathways were classified into hierarchical categories according to the KEGG pathway website (https://www.kegg.jp/kegg/pathway.html, accessed on 11 November 2021), as shown in Figure 7D.

At 0 d, compared to LD, KEGG cluster analysis showed that the DAPs in ND were highly enriched in protein processing in the endoplasmic reticulum, while those in HD were highly enriched in protein synthesis and shear functions, as well as in photosynthesis. A similar enrichment pattern was observed in HD vs. ND, but with higher abundances. At 20 d, the DAPs in ND vs. LD were highly enriched in oxidative phosphorylation, arachidonic acid metabolism, and photosynthesis, while those in HD vs. LD were highly enriched in amino acid metabolism, lipoid metabolism, sulfur metabolism, and organic synthesis. The DAPs in HD vs. ND were highly enriched in sugar metabolism, protein processing, and circadian rhythm. At 40 d, the DAPs in ND vs. LD were highly enriched in amino acid metabolism, while the DAPs in HD vs. LD were highly enriched in amino acid metabolism; protein and lipid metabolism; glycan degradation; and response to stress. A similar enrichment was observed in HD vs. ND, except that antioxidant proteins were also significantly enriched.

### 2.6. Abundance of Proteins in the Photosynthetic Apparatus

Photosynthesis-related proteins are among the most important DAPs under low-light stress [13,14,20]. In this study, a further KEGG analysis showed that the up-regulated photosynthetic DAPs were highly enriched in antenna proteins and the C3 pathway. The antenna proteins were mainly in LD and ND, while C3 were mainly in HD. However, the down-regulated photosynthetic DAPs were highly enriched in PSII, and the number of DAPs increased with plant density at leaf senescence (Appendix A). In other words, the photosynthetic response to low light was associated with changes in proteins related to photosynthesis, especially light-dependent photosynthesis.

Of the proteins related to light-dependent photosynthesis, we identified 33 significantly DAPs among the nine experimental groups (three densities × three growth stages; *p* < 0.05) (Figure 8B), including eight proteins related to the light-harvesting chlorophyll complex (LHC), nine proteins related to PSII, seven proteins related to cytochrome *b*_6_*f* (Cyt*b*_6_*f*), six proteins related to PSI, and three proteins related to ATPase.

Compared to LD, the abundance of LHCB4 and LHCB5 was higher in HD at 0 d, and the abundance of LHCB1, LHCB2, and LHCB4 was higher in ND at 40 d (Figure 8B). Moreover, the abundance of LHCA1, LHCA3, and LHCA4 increased with the increased planting density at 0 d. These results were consistent with the ABS/RC patterns (Figure 5). In PSII, the identified DAPs were related to the oxygen evolution complex (OEC), which is an important component of PSII and is responsible for the cleavage and oxidation of H_2_O [52]. At 0 d, the abundances of Psb28, PsbF, PsbL, and PsbP increased with the increased planting density; the abundance of Psb27 was higher in HD than in LD, and the abundance of PsbQ was higher in ND than in LD. However, the abundances of PsbE, PsbF, and PsbQ were higher in ND than in LD at leaf senescence (20 d). These results indicate that the effect of abiotic stress on OEC-related proteins depends on the type and the duration of stress.

In cytochrome *b*_6_*f* (Cyt*b*_6_*f*), the abundances of PetA, PetD, and PetF were higher in ND than in LD at 0 d, while the abundances of PetD, PetF, and PetJ were higher in HD than in LD. In PSI, the abundances of PsaC, PsaG, PsaJ, and PsaN were higher in ND than in LD at 0 d, while the abundances of PsaG, PsaH, and PsaJ were higher in HD than in LD (Figure 8B). Moreover, the levels of PsaG were higher in ND than in LD at 20 d, and the levels of PsaN were higher in HD than in LD at 40 d. In addition, proton (H^+^) translocation from the chloroplast stroma into the lumen establishes a transmembrane pH gradient that serves as a proton-motive force to drive ATP synthesis [53] (Figure 8A). The levels of ATPF0B and ATP1D were higher in ND than in LD, while ATPF0A and ATPF1D were more abundant in HD than in LD (Figure 8B). However, the levels of all ATPase-related proteins were lower in ND and HD than in LD at leaf senescence.

## 3. Discussion

### 3.1. Low-Light Stress Induced by Dense Planting Affects Leaf Physiology and Photosynthetic Gas Exchange

Photosynthesis contributes to plant growth and crop yield [54,55]. Plant density can be altered to optimize canopies, allowing plants to capture more light energy and increase canopy photosynthetic capacity [56]. In maize, however, low-light stress often arises as a result of self-shading as plant density increases. Low-light stress reduces the leaf photosynthetic rate, thus reducing photosynthetic product levels and biomass accumulation [3,57], mainly as a result of changes in leaf physiology and anatomy, chloroplast ultrastructure, and photosynthetic characteristics [7,58,59]. Here, we found that leaf mass per area (LMA), an important parameter of leaf structure that is closely associated with the light environment and with leaf photosynthesis [60], significantly decreased with increasing planting density at leaf senescence (Figure 1A), similar to previous findings [56]. Efficient light energy capture is of paramount importance for plants growing in dense stands [61,62], and lower LMA is a key modification to enhance light harvesting [56,63]. This could explain why LMA was lower in HD than in ND and LD maize plants (Figure 1A).

Thinner leaves with lower LMA often have greater leaf photosynthetic capacity [64,65]. In our study, ND and HD leaves had a lower LMA and a higher A_max_ than in LD leaves at 30 and 40 d after anthesis; however, the opposite was true at 0, 10, and 20 d after anthesis (Figure 1A; Table 1). These differences may be due to changes in leaf chlorophyll content (Chl (*a* + *b*)) over time, as Chl (*a* + *b*) and LMA are important factors determining A_max_ during leaf senescence [56]. We found that Chl (*a* + *b*) followed a similar trend to LMA; however, the differences between planting densities were smaller (Figure 1A,B). Therefore, we suggest that the photosynthetic capacity under different plant density treatments is the result of coordination between LMA and Chl (*a* + *b*) as the leaves age.

Moreover, leaf N content was significantly lower in ND and HD leaves than in LD leaves (Figure 1C), similar to the results of previous studies on several C3 and C4 species [56,66,67]. Chlorophyll is destroyed as N is remobilized from old leaves and translocated to new leaves after anthesis, and leaf Chl (*a* + *b*) content also affects light energy capture [68,69]. Therefore, the lower leaf N content after anthesis at higher planting densities could reduce the light-harvesting capacity. This was reflected in the energy absorbed by active RCs (ABS/RC) (Figure 5) and the expression levels of antenna proteins (Appendix A). In cotton (*Gossypium hirsutum*), lower light energy capture at key growth periods under low- and high-density planting leads to a lower light utilization efficiency and thus lower photosynthetic production [56]. However, the reduction in absorbed photosynthetically active irradiance is, to a certain extent, compensated for by an increase in light use efficiency, thereby decreasing the difference in photosynthetic productivity between shaded (corresponding to high density in this study) and non-shaded (corresponding to low density in our study) plants [70]. In this study, dense planting treatments led to a decrease in the light compensation point (LCP) and dark respiration rate (R_d_) but an increase in the apparent photosynthetic quantum yield (AQE) (Table 1), similar to what was found in other studies [3,71]. These results suggest that maize plants in dense plantings can adapt to low light by increasing light utilization efficiency and reducing photosynthates consumption. Measuring the changes of photosynthetic gas exchange confirmed that the value of P_n_ was significantly higher in ND and HD leaves than that in ND leaves under low-light conditions at the middle and later growth stages (Figure 2).

### 3.2. Low-Light Stress Induced by Dense Planting Affects Photosynthetic System Performance

Low-light or shade conditions impair the light and dark reactions in photosynthesis and thus reduce photosynthetic productivity [72,73,74]. In this study, we focused on the effect of low-light stress induced by dense planting on the light reaction in photosynthesis. Photosynthetic systems, including PSII and PSI, play important roles in light energy absorption, transformation, and transmission and are highly sensitive to changes in light intensity [75,76,77]. Chlorophyll *a* fluorescence can be used to evaluate plant photosystem performance under stress conditions. Our study showed that plant density significantly affected PSII performance, and the effects were visible in the variable fluorescence curves (V_t_, Figure 3A–C) as well as in the relative variable fluorescence curves (ΔV_t_, Figure 3A–C). There were two clear bands in fluorescence intensity in ΔV_t_, namely a ΔK (at ~300 μs) and ΔJ peak (at ~2 ms). The ΔK bands occurring during the O–J transient are usually due to an imbalance in electron transfer reactions at donor and acceptor sides of PSII [78,79] and are associated with the uncoupling of the oxygen-evolving complex (OEC) in PSII [80]. A positive ΔK band is usually explained as the result of a retardation of electron donation from the OEC to oxidized chlorophyll, leading to increased PSII reaction center (P680^+^) concentration, which can effectively quench the excited antenna chlorophylls [31]. This is reflected in an increased value of W_k_ during the O–J transient [75]. In addition, the ΔJ bands occurring during the O–J transient are usually associated with the accumulation of Q_A_^−^ [81]. A positive ΔJ band is usually explained as a consequence of retardation of electron acceptance from Q_A_ to Q_B_ [75], leading to an inhibition of the Q_A_^−^ reoxidation that effectively inhibits transfer electrons to the dark reactions. This inhibition is reflected in an increased value of V_j_ during the O–J transient [76,81]. In this study, the ΔK and ΔJ bands decreased in the order LD > HD > ND within each growth stage (Figure 3A–C), as did W_k_ and V_j_ (Figure 5), suggesting that the performances of the PSII donor and acceptor side were impaired by LD at each growth stage, compared to ND and HD. This is consistent with a previous study that found a reduced rate of electron donation to the PSII RC due to impaired OEC [78].

The leaf pipeline model has been widely applied to describe environmental and/or anthropogenic pressure on plants [30,31,81]. In our study, a phenomenological energy flux model revealed that the absorbed energy (ABS/CS), trapped energy (TR/CS), energy transfer (ET/CS), and energy dissipation (DI/CS) per cross-section were higher at ND and HD than at LD from 20 to 40 d after anthesis (Figure 4). In contrast, the absorbed energy per active RC (ABS/RC) was lower at ND and HD than at LD (Figure 5). This indicates that either the number of active RCs increased or the apparent antenna size decreased [81]. In our study, the active RCs per excited cross section (RC/CS) increased (Figure 4). This indicates that more excitons were transferred to the plastoquinone, which likely alleviated all trapped energy was dissipated as heat [82,83]. Indeed, the values of φD_o_ and DI_o_/RC in ND and HD were lower than those in LD leaves (Figure 5). Moreover, the decrease in ABS/RC (or increase in the active RCs) was accompanied by a decrease in trapping per active RC (TR_o_/RC), which is an indicator of impairment to the OEC due to stress and is calculated similarly to W_k_ (Table 2). However, the electron transport flux per active RC (ETo/RC) and transport from Q_A_^−^ to the PSI electron acceptors (RE_o_/RC) were higher in ND and HD than in LD leaves (Figure 5). These results indicate that the maize plants grown in ND and HD conditions adjusted their ratio of energy distribution by increasing their electron transfer rate and reducing energy dissipation to increase their ratio of light energy utilization. These findings were supported by an increase in the parameters representing quantum yield and efficiency (ψ_o_, φE_o_, φR_o_, and δR_o_) (Figure 5). The increased values of these parameters in ND and HD suggest increased electron transport from plastoquinone A to B and increased levels of PSI electron acceptors [84,85], which could reduce the electron flow to O_2_ during photosynthetic and respiratory processes. In other words, the improvements to intersystem electron transport and the PSI end electron acceptor in ND and HD treatments likely decreased the accumulation of reactive oxygen species (ROS), alleviating damage to the thylakoid membranes under low-light stress induced by dense planting [77,86].

The performance index on an absorption basis (PI_abs_), which is calculated from several parameters in the OJIP fluorescence induction curves, is often used to quantify the overall performance of PSII. The PI_abs_ was significantly higher in ND and HD leaves than in LD leaves at each growth stage, suggesting that the photochemical efficiency of PSII increased under low-light stress induced by dense planting, similar to previous findings [3]. Furthermore, PI_total_ reflects the performance from the PSII electron donor side to the reduction of the PSI end electron acceptors (RE), which involves several electronic transport processes, such as φP_o_, ψ_o_, and δR_o_ (Table 2). In this study, ψ_o_ and δR_o_ were significantly higher in ND and HD leaves than in LD leaves at 0, 20, and 40 d after anthesis; however, the maximum yield of primary photochemistry of PSII (φP_o_) was similar among the three plant densities at each growth stage. This led to higher PI_total_ in ND and HD leaves. These results suggested that electron transport was improved from PSII to PSI, which may have resulted from the higher activity of PSI [88]. Indeed, ΔI/Io, which was calculated from 820-nm reflection as an index of the content of active PSI RCs [89,90,91], was significantly higher in ND and HD than that in LD leaves at each growth stage (Figure 5). Consequently, the higher PSI and PSII activity (as shown in the number of RCs in Figure 4) could have promoted increased coordination between PSII and PSI (Φ_(PSI/PSII)_, Figure 5) in ND and HD leaves at each growth stage, which in turn could have ensured efficient intersystem electron transport.

### 3.3. Low-Light Stress Induced by Dense Planting Affects Leaf Proteomics

Proteomics provides insights into gene regulation and has been widely used to study plant responses to low-light stress [13,19,20]. The proteins that are differentially expressed in response to low-light stress are involved in many cellular functions and differ among crop cultivars. For example, proteins related to the photosynthetic electron transport chain and stress/defense/detoxification play an important role in the low-light adaptation of maize leaves [14], while proteins related to porphyrin and chlorophyll metabolism, photosynthesis-antenna, and carbon fixation play an important role in the low-light adaptation of soybean leaves [20]. Moreover, the proteins that are differentially expressed in the same crop can also vary with growth period [14]. In this study, compared to LD, ND and HD induced an increase in the abundance of ribosome and phosphorylation proteins at 0 d after anthesis, of photosystem and electron transport chain related proteins at 20 d after anthesis, and of cell wall, respiratory chain, and ATP synthase complex proteins at 40 d after anthesis (Figure 7). These changes may explain why the values of A_max_ were significantly higher in ND and HD than in LD leaves from 20 to 40 d after anthesis (Table 1). Similarly, a previous study showed that a higher photosynthetic capacity of leaves was maintained under shaded conditions by an increase in the abundances of proteins related to photosynthesis [19,92].

The proteomic results in this study confirmed that the chloroplast proteins were most affected by low-light stress induced by high planting density (Figure 6D) [5,6,13]. Of these proteins, photosynthesis-related proteins are the most important [14,19,20]. During photosynthesis, the light-harvesting capacity is affected by Chl (*a* + *b*), mainly owing to changes at the light-harvesting complex (LHC) level [93]. At 0 d after anthesis, the expression of LHC-related proteins was higher under low-light conditions induced by ND and HD than that in LD leaves (Figure 8B), similar to previous findings [19,20]. The increased abundance of LHC in dense planting may have provided more excitation energy for the reaction center, requiring the OEC to provide more electrons and thereby driving the increase in its related protein levels, as shown by the changes in PsbO, PsbQ, and PsbP (Figure 8A,B). Although low-light stress can lead to the degradation of D1 protein (PsbA) on the acceptor side of PSII and thus inhibit electron transport from PSII to PSI [94], we did not observe any significant differences in PsbA under dense planting in this study (Figure 8B; Appendix A). This indicates that the degree of stress induced by dense planting is not enough to cause PSII photoinhibition and D1 protein degradation, which was confirmed by the abundances of PSII-related proteins (Figure 8B). It is noteworthy that the values of V_j_, representing the performance of the acceptor side of PSII [76], were significantly higher in ND and HD leaves than in LD leaves at each growth stage (Figure 5). These differences may result from the increase in OEC-related proteins leading to a balance in electron transfer reactions at the donor and acceptor sides of PSII and the higher PSI activity resulting in an efficient electron transport from PSII to PSI, rather than the changes of D1 protein [78,79,81,88].

Electrons are continuously shuttled through pheophytin (Phe) and the plastoquinone molecular Q_A_ and Q_B_, and then transported to the chloroplast cytochrome *b*_6_*f* complex (Cyt*b*_6_*f*), which is located between PSII and PSI (Figure 8 A). The expression of Cyt*b*_6_*f*-related proteins can be increased under stress conditions [19,95]. This was observed in this study: The abundances of PetA, PetD, PetF, and PetJ were higher in ND and HD leaves than in LD leaves (Figure 8B). The higher abundances of these proteins may constitute a coordinated strategy that ensures efficient electron transport and maintains contact with upstream metabolism under dense planting [53,96]. However, opposite results have also been reported. For example, low-light stress has been found to significantly decrease the activity and/or abundance of Cyt*b*_6_*f* [5,6,13]. It is possible that the degree of stress in this study was not sufficient to cause photodamage in maize leaves, which was also suggested by the levels of OEC-related and D1 proteins (Figure 8B).

In the next step of photosynthesis, electrons are transported from the Cyt*b*_6_*f* complex to PSI by plastocyanin (PC). After activation through a second light reaction (P700) of PSI, electrons are further transported by two pathways: non-cyclic electron transport and cyclic electron transport. In the non-cyclic pathway, ferredoxin (Fd) transfers a single electron to the flavoprotein ferredoxin NADP^+^ reductase (FNR), which then transfers two electrons to NADP^+^ and H^+^ to produce NADPH that is used in the Calvin–Benson cycle (Figure 8A). However, in the cyclic pathway, Fd transfers a single electron back to plastoquinone (i.e., Cyt*b*_6_*f*), which can participate in cyclic electron transport with PSI and generates ATP without the accumulation of NADPH, playing an important role in photoprotection (such as non-photochemical quenching, NPQ) (Figure 8A). In this study, the levels of PSI-related proteins, such as PsaC, PsaG, PsaH, and PsaJ, were higher in ND and HD leaves than in LD leaves (Figure 8B), similar to previous findings [19]. Although the higher abundances of Cyt*b*_6_*f* and PSI proteins may increase cyclic electron transport [19,97,98,99], there were no significant differences in the abundance of PsbS (the protein regulating NPQ) [100,101,102] among the three plant densities (Figure 8B; Appendix A), suggesting that low light induced by dense planting increased non-cyclic electron transport.

ATPase is the key enzyme of photophosphorylation; it plays an important role in the conversion of energy in photosynthesis [103] and is sensitive to abiotic stresses, such as salt, drought, and low-light/shade conditions [19,104,105,106,107]. In this study, the abundances of ATPase-related proteins, such as ATPFF0A, ATPF0B, and ATPF1D, were higher in dense planting than in LD plantings (Figure 8B). Increased ATPase and enhanced linear electron transport under dense planting can increase ATP production, which can promote the Calvin–Benson cycle because Rubisco activation requires energy released by ATP hydrolysis, while Rubisco is the rate-limiting enzyme of the Calvin–Benson cycle [108]. This hypothesis was supported by the fact that the values of P_n_ and A_max_ in the later growth stages were significantly higher in ND and HD leaves than in LD leaves (Figure 2; Table 1).

In conclusion, our results showed that low-light stress induced by dense planting increased the abundances of proteins related to the light reactions in photosynthesis, including seven proteins related to the LHC (LHCB1, LHCB2, LHCB4, and LHCB5 and LHCA 1, LHCA3, and LHCA4), four proteins related to PSII (Psb28, PsbF, PsbL, and PsbP), four proteins related to Cyt*b*_6_*f* (PetA, PetD, PetF, and PetH), three proteins related to PSI (PsaC, PsaG, and PsaJ), and three proteins related to ATPase (ATPF0A, ATPF0B, and ATPF1D). The main function of these proteins is to facilitate photosynthetic electron transport. As a result, the parameters obtained from chlorophyll fluorescence were higher in ND and HD leaves than in LD leaves, including φE_o_, ψ_o_, δR_o_, and φR_o_. In other words, the improvement to intersystem electron transport and PSI end electron acceptors under ND and HD treatments likely decreased the damage to thylakoid membranes caused by low-light stress induced by dense planting. This hypothesis was supported by the fact that maize plants under dense planting can adapt to low light by increasing their light utilization efficiency and reducing consumption of photosynthates, which was reflected in the higher AQE and lower R_d_ and LCP. In contrast to the reduced nutrient and water uptake and utilization efficiency caused by the changes of root phenotype, light utilization efficiency significantly increased under low light, although densely planted maize showed substantial alterations in leaf morphological and physiological traits. In addition, our results showed the potential to utilize chlorophyll fluorescence parameters, as well as the related proteins, as indicators of plant light stress, which could provide technical support for developing cultivation conditions and maize varieties suited to intensive agriculture.

## 4. Materials and Methods

### 4.1. Experimental Design and Field Management

The field experiment was conducted at Shandong Agricultural University Experimental Farm in Tai’an, Shandong Province, China (36°10′19″ N, 117°9′03″ E) during the 2017 maize growing season. This region has a semi-humid, temperate continental monsoon climate. The mean air temperature during the maize growth period was 24.8 °C, and the average precipitation was 485.1 mm. The soil type at the experimental site is a brown loam, and the nutrient status of the upper 20 cm prior to seeding was as follows: organic matter, 11.4 g kg^−1^; total nitrogen (N), 0.71 g kg^−1^; available phosphorous (P), 25.6 mg kg^−1^; available potassium (K), 107.2 mg kg^−1^.

The summer maize hybrid Zhengdan 958 (a high-yield and density-tolerant variety grown extensively in the North China Plain) was selected for the experiment. Maize seeds were planted with hand planters at three densities: low density (LD; 22,500 plants hm^−2^), normal density (ND; 67,500 plants hm^−2^), and high density (HD; 90,000 plants hm^−2^), with three replicate plots for each treatment. Each plot was 30 m long × 9 m wide and consisted of 15 rows with 0.6 m row spacing. Plants were placed 74, 24.5, and 18.5 cm apart within a row to achieve the LD, ND, and HD planting densities, respectively. The six rows in the middle of each plot were used in collecting experimental data. Phosphorus (P_2_O_5_) and potassium (K_2_O) fertilizer were applied at a rate of 150 kg hm^−2^ in a single dose to a depth of 5 cm in bands between rows at the three-leaf stage. Urea (N, 46%) fertilizer was split at a rate of 225 kg hm^−2^ in two doses, 40% at the three-leaf stage, and the rest at the large-bell stage, using a similar method as that used for phosphorus and potassium fertilizer. Irrigation, weeds, disease, and insect pests were controlled in each treatment.

### 4.2. Leaf Mass per Area (LMA), Leaf Chlorophyll Content (Chl (a + b)), and Leaf N Content

LMA was calculated by dividing the leaf dry mass by the leaf area. At 0, 10, 20, 30, and 40 d after anthesis, five representative plants of each treatment group were selected for measurement of green leaf length (L) and maximum leaf width (W) (leaf area = L × W × 0.75). Fresh discs (~0.1 g) were removed from each leaf and placed in 80% (*v*/*v*) acetone for 24 h of dark adaptation at room temperature, and leaf chlorophyll content was spectrophotometrically (UV2450, Shimadzu Co., Tokyo, Japan) calculated according to the method of Agron [109]. The remaining fresh leaves were then killed in an oven at 105 °C for 30 min and dried at 70 °C to constant weight. Dried leaf samples (0.2 g) were digested in H_2_SO_4_–H_2_O_2_ solution following the micro-Kjeldahl method [110].

### 4.3. Gas Exchange Parameter Measurement

Three representative plants of each treatment were tagged for the measurement of photosynthetic gas exchange. At 0, 10, 20, 30, and 40 d after anthesis, the net photosynthetic rate (P_n_), stomatal conductance (G_s_), and intercellular CO_2_ concentration (C_i_) under high light (HL, 1600 μmol m^−2^ s^−1^) and low light (300 μmol m^−2^ s^−1^) were established with a portable photosynthesis apparatus (CIRAS-3, PP system, Amesbury, MA, USA) in the late morning (09:00–11:00). The rapid response of A_net_ to irradiance corresponded to the following light intensities: 0, 100, 200, 300, 500, 800, 1000, 1200, 1400, and 1600 μmol m^−2^ s^−1^. The leaf temperature, relative humidity, and CO_2_ level in the leaf chamber were set at 25 °C, 65%, and 400 μmol mol^−1^, respectively. A non-rectangular hyperbola was fit to the A-PPFD response curve data according to the method of Thornley [111] using the nonlinear regression procedure in SPSS (version 17.0, SPSS Inc., Chicago, IL, USA) to calculate the apparent quantum efficiency (AQE):$$A=\frac{\mathrm{AQE}*\mathrm{PPFD}+A_{max}-\sqrt{(AQE*PPFD+Amax{)}^{2}-4*k*PPFD*Amax}}{2*k}-R_{d}$$
where PAR is the photosynthetically active radiation, A_max_ is the maximum gross photosynthetic rate, *k* is the scaling constant for the curve, and R_d_ is the dark respiration rate.

### 4.4. Chlorophyll a Fluorescence Induction Transient Analysis and 820-nm Reflection Curves of Leaves

Chlorophyll *a* fluorescence (ChlF) induction transient analysis was conducted using the same leaves employed for gas exchange measurements with a *HandyPEA* fluorimeter (Handy Plant Efficiency Analyzer, Hansatech Instruments Ltd., King’s Lynn, Norfolk, UK) according to the method presented by Schansker et al. [112] with slight modifications. The leaves were dark-adapted using leaf clips for 15 min before fluorescence measurements. The dark-adapted leaf samples were illuminated with a saturating light pulse of 3500 μmol m^−2^ s^−1^ for 1 s. At least 15 measurements were performed for each treatment. The ChlF intensity was recorded in arbitrary units and then transformed into relative variable fluorescence (V_t_) in relative units by double normalization to F_0_ (minimum fluorescence) and F_m_ (maximum fluorescence): V_t_ = (F_t_ − F_0_)/(F_m_ − F_0_). Then, the V_t_ values of the leaves at 0 d after anthesis were subtracted from the values of the leaves at 10, 20, 30, and 40 d after anthesis at the same planting density. This resulted in induction differential curves: ΔV_t_ = (F_t_ − F_o_)/(F_m_ − F_o_) − V_t, 0 d_ [113]. The basic fluorescence intensities determined at 20 μs, 300 μs, 2 ms, 30 ms, and F_m_ were used to calculate the JIP-test parameters that are shown in Table 2. Then, the light absorption curves at 820 nm were measured using M-PEA (Hansatech, King’s Lynn, UK) after the leaves were dark-adapted for 15 min [114]. The relative amplitude of 820-nm light absorption (ΔI/I_o_ = (I_o_ − I_m_)/I_o_) was used as an index of the relative content of active PSI reaction centers [89,90,91].

### 4.5. TMT-Based Quantitative Proteomics Analysis

According to previous studies, we chose 0, 20, and 40 d after anthesis as three distinct phases during which to determine leaf proteins abundances changes during the stages of grain yield formation. Samples were collected after physiological measurement in each stage. The middle portions of the leaves (with the veins removed) were collected, frozen in liquid nitrogen, and stored at −80 °C prior to proteomics analysis.

Proteins were extracted from frozen leaves (three planting densities × three growth stages) with three biological replicates. First, the samples were ground into a powder in liquid nitrogen, and then proteins were extracted in lysis buffer. The protein concentrations were quantified using a 2D Quant kit (GE Healthcare Bioscience, Shanghai, China), and the quality of extracted proteins was detected by sodium dodecyl sulfate polyacrylamide gel electrophoresis (SDS-PAGE). Then, 10 protein samples, including an internal standard (ISTD: 50 μg of each protein sample mixed equally), were digested with trypsin. Next, the digested peptides were desalted, vacuum-dried, and labeled using a TMT 10-plex Isobaric Label Reagent Set (Thermo Fisher Scientific, San Jose, CA, USA) following the manufacturer’s instructions (Appendix A). Finally, the samples were fractionated by high-pH reverse-phase high-performance liquid chromatography (HPLC) fractionation, which was performed by liquid chromatography–tandem mass spectrometry (LC–MS/MS) analysis. Further details of the TMT-based proteomics analysis methods are described in Appendix A.

The resulting raw LC–MS/MS spectra were analyzed using the MaxQuant search engine (version 1.4.1.2) against the Uniprot database. Search parameters were as follows: monoisotopic mass; trypsin as cleavage enzyme; two max missed cleavages; TMT 10-plex (N-term), TMT 10-plex (K), and carbamidomethylation of cysteine as fixed modifications; and oxidation of methionine as variable modifications. The mass error was set to 10 ppm for precursor ions and 0.02 Da for fragmentation. The false discovery rate (FDR) was set to 1% for peptide and protein identification. For the database search, the length of the shortest peptide was set to seven amino acid residues. All other parameters used in MaxQuant were default settings.

The TMT reporter ion intensity was used for protein quantification. At least one unique peptide was required per protein from all three replicates in the follow up quantification analysis. The protein ratios in each replicate were quantified according to the summed intensity of the matched spectra. An arithmetic mean value of ratios of different TMT reporters in the three biological replicates was used as the quantitative result of each treatment. Since the measurement of protein amount may be inaccurate, the digestion may have different efficiencies and post-digestion purification may have different peptide recoveries, the actual ratio of labeled peptide samples thus may not be 1:1. Typically, ratio variation of 1.5-fold or less is acceptable and can be normalized latter, but if it too large, the channels with lower reporter ion intensities will be skewed towards zero [115]. To avoid the changes of DAPs caused by sampling or experimental error, and the loss of important proteins, proteins with a *p*-value < 0.05 (Student’s *t*-test) and a fold-change >1.30 or <0.77 were considered DAPs in this study [116,117,118,119]. The gene ontology (GO) database (http://geneontology.org/, accessed on 22 October 2021) was used to determine DAPs categories. The Kyoto Encyclopedia of Genes and Genomes (KEGG) database (https://www.kegg.jp/kegg/pathway.html, accessed on 11 November 2021) was used to annotate protein pathways. GO and KEGG enrichment analyses were performed using the Fisher’s exact test, and FDR correction for multiple testing was also performed. Furthermore, we used the online tools (https://bioinfogp.cnb.csic.es/tools/venny/index.html, accessed on 18 February 2022) to analyze the overlap of proteins between different treatments to make the Venn diagrams.

### 4.6. Statistical Analysis

All statistical analyses were performed with SPSS 17.0 (SPSS Institute Inc., Chicago, IL, USA). One-way ANOVA was used to assess the effects of planting density on plant physiology data presented, with the least significant difference (LSD) tests at *p* ≤ 0.05. Plant physiology data were mapped using Sigmaplot 10.0 (Systat Software Inc., San Jose, CA, USA) and Origin 2017.

## Figures and Tables

**Figure 1 ijms-23-03015-f001:**
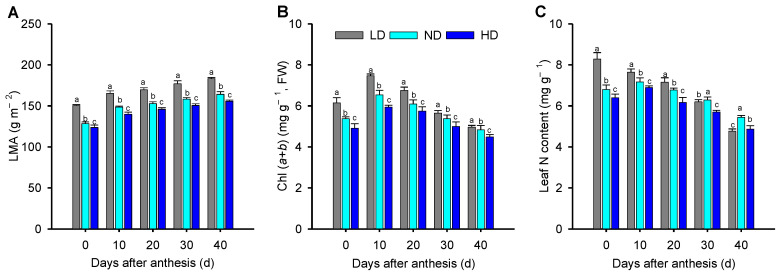
Maize leaf physiological parameters under low (LD), normal (ND), and high density (HD) planting at 0, 10, 20, 30, and 40 d after anthesis. (**A**) Leaf dry mass per area (LMA), (**B**) leaf total chlorophyll content (Chl (*a* + *b*)) and (**C**) leaf nitrogen content. Means ± SD, *n* = 5. Different lowercase letters (a, b, c) denote statistical differences by LSD test (*p* ≤ 0.05) between different treatments.

**Figure 2 ijms-23-03015-f002:**
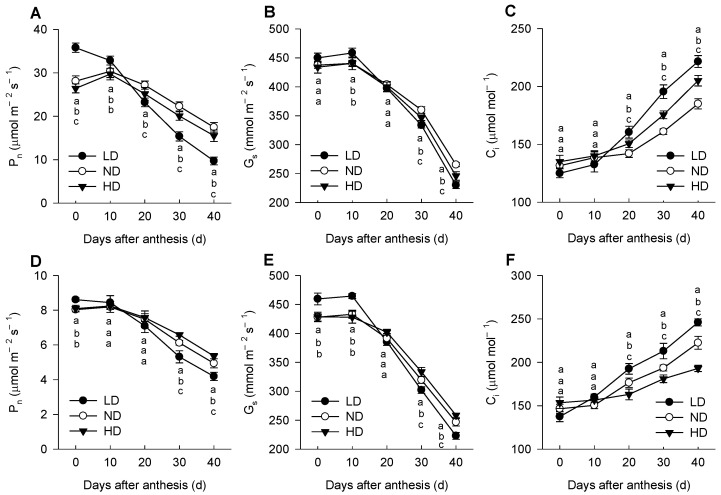
Changes in net photosynthetic rate (P_n_, (**A**,**D**)), stomatal conductance (G_s_, (**B**,**E**)), and intercellular CO_2_ concentration (C_i_, (**C**,**F**)) of maize leaves under high-light (HL, 1600 μmol m^−2^ s^−1^; (**A**–**C**)) and low-light (LL, 300 μmol m^−2^ s^−1^; (**D**–**F**)) conditions. Means ± SD, *n* = 5. Different lowercase letters (a, b, c) denote statistical differences by LSD test (*p* ≤ 0.05) between different treatments. LD, low density; ND, normal density; HD, high density.

**Figure 3 ijms-23-03015-f003:**
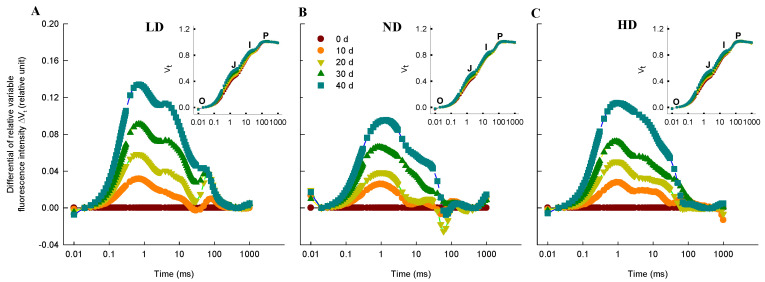
Chlorophyll *a* fluorescence (ChlF) induction transient curves of maize leaves. (**A**–**C**) Effects of planting density ((**A**), low density, LD; (**B**), normal density, ND; (**C**), high density, HD) on differential plots of relative ChlF (ΔV_t_ = (F_t_ − F_o_)/(F_m_ − F_o_) − V_t,0d_) in the leaves of maize at 0, 10, 20, 30, and 40 d after anthesis. For ΔV_t_ analysis, the fluorescence of leaves at 0 d after anthesis was used as a reference and set to 0. For the insert plot in (**A**–**C**), the chlorophyll a fluorescence (ChlF) transient OJIP kinetics curves, O, J, I, and P, represent the fluorescence intensity at 20 μs, 2 ms, 30 ms, and 500 ms, respectively. Values are means (*n* = 9).

**Figure 4 ijms-23-03015-f004:**
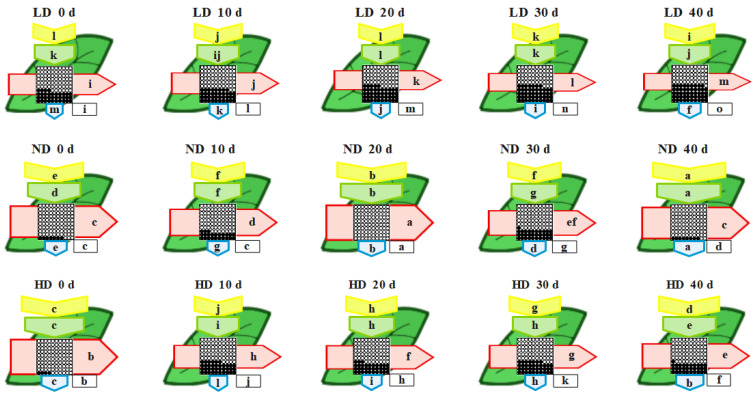
Leaf models showing the phenomenological energy fluxes per excited cross section (CS) of maize leaves grown under low (LD), normal (ND) and high density (HD) planting at 0, 10, 20, 30, and 40 d after anthesis. Each relative value is the mean (*n* = 9), and the width of each arrow corresponds to the intensity of the flux. ABS/CS, approximate absorption flux per CS (yellow arrows); TR/CS, trapped energy flux per CS (green arrows); ET/CS, electron transport flux per CS (red arrows); DI/CS, dissipated energy flux per CS (blue arrows); RC/CS, percent of active/inactive reaction centers (circles inscribed in squares). White circles inscribed in squares represent reduced Q_A_ reaction centers (active), black circles represent non-reducing Q_A_ reaction centers (inactive), and 100% of active reaction centers represent the highest mean value observed during the three measured stages. Means followed by the same letter (a–m) for each parameter are not significantly different from each other using the LSD test (*p* ≤ 0.05). Letters are inscribed into arrows, except for RC/CS where they are placed in a box in the lower right corner of the square with circles. LD, low density; ND, normal density; HD, high density; 0, 20, and 40 d, days after anthesis.

**Figure 5 ijms-23-03015-f005:**
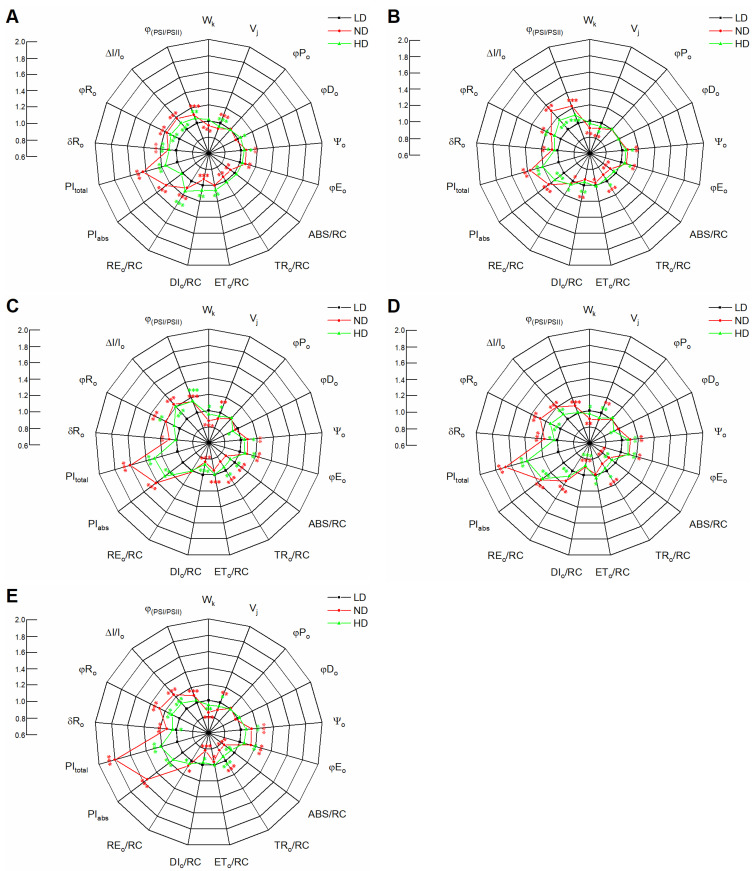
Spider plots showing changes in JIP-test parameters in maize leaves planted in low density (LD), normal density (ND), and high density (HD) and measured at 0 (**A**), 10 (**B**), 20 (**C**), 30 (**D**), and 40 d (**E**) after anthesis. Values are means (*n* = 9). Asterisks (*, **, and ***) denote significant differences between different planting densities according to Fisher’s LSD test at *p* ≤ 0.05, *p* ≤ 0.01, and *p* ≤ 0.001, respectively. Significant differences between low and normal (red) or high (green) density treatment are denoted by asterisks of different colors.

**Figure 6 ijms-23-03015-f006:**
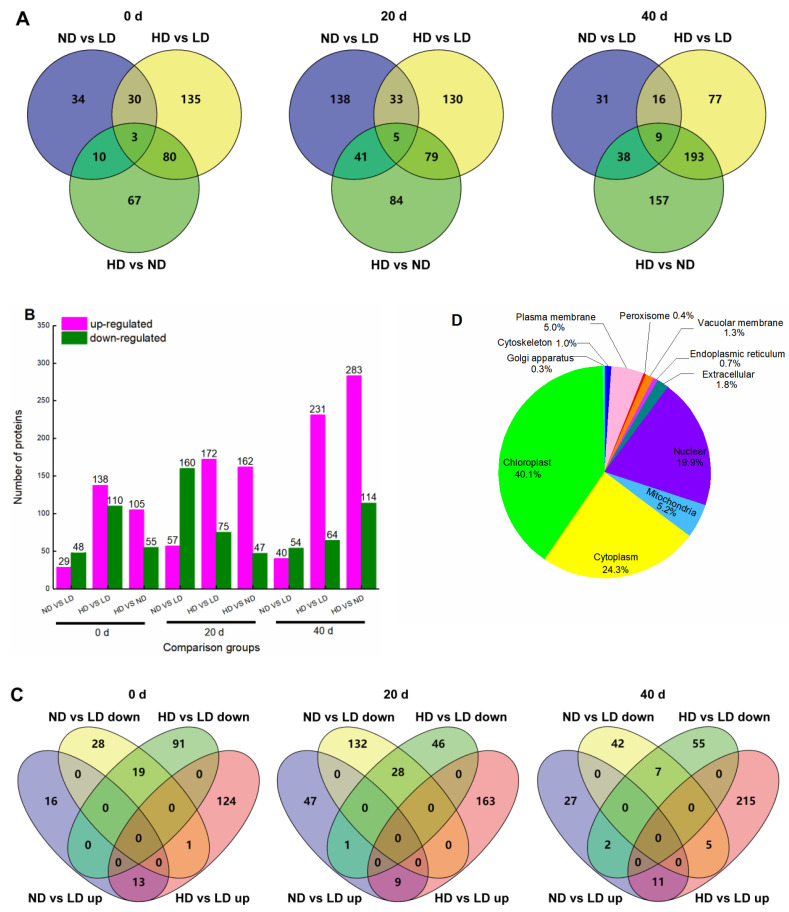
Overview of protein accumulation**.** (**A**) Venn diagrams depicting the overlap of differentially abundant proteins (DAPs) between different treatments at 0, 20, and 40 d after anthesis. (**B**) Bar chart showing the number of up- and down-regulated DAPs in each of the comparison groups at 0, 20, and 40 d after anthesis. Magenta color indicates up-regulated proteins, and green color indicates down-regulated proteins. (**C**) Venn diagram showing the distribution of up- and down-regulated DAPs in ND vs. LD and HD vs. LD. (**D**) Localizations of identified proteins. LD, low density; ND, normal density; HD, high density.

**Figure 7 ijms-23-03015-f007:**
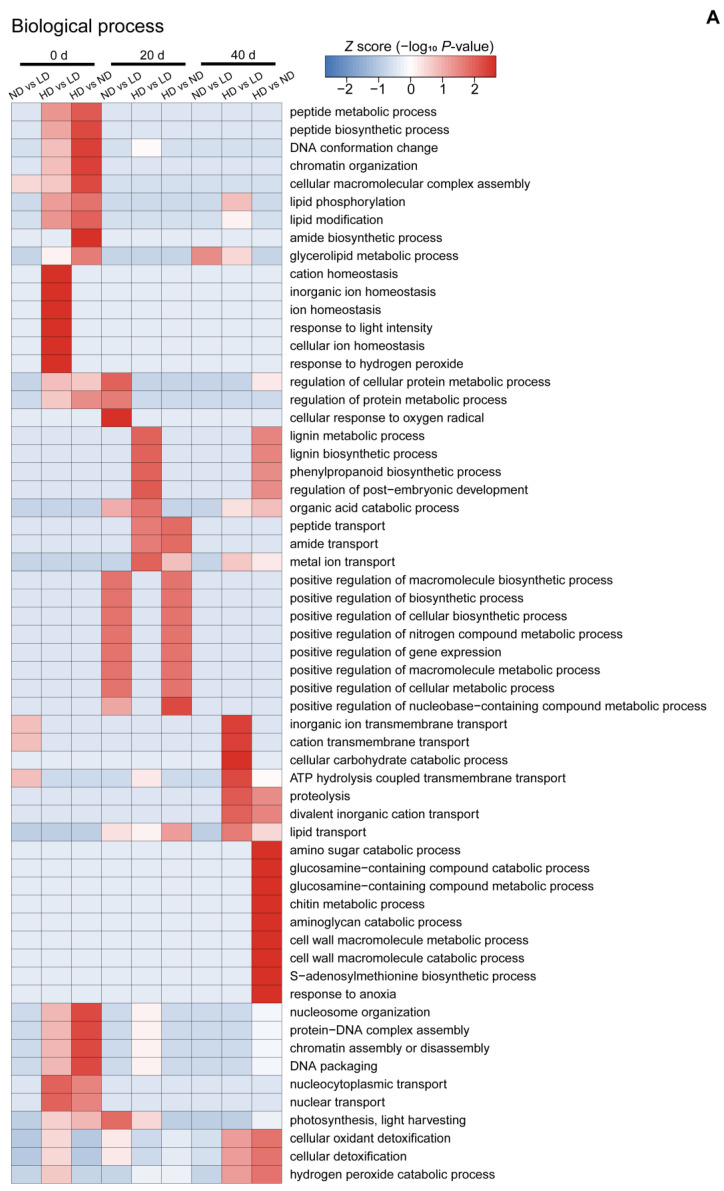
Heatmaps obtained from GO and KEGG pathway analysis comparing the differentially abundant protein (DAP) expression patterns under different conditions. (**A**) Biological process analysis; (**B**) cellular component analysis; (**C**) molecular function analysis; and (**D**) KEGG pathway analysis. Colored scales of the *Z* score (–log_10_ *p*-value) are shown; proteins that accumulated at high levels are in red, and proteins with low accumulation levels are in blue. LD, low density; ND, normal density; HD, high density; 0, 20, and 40 d, days after anthesis.

**Figure 8 ijms-23-03015-f008:**
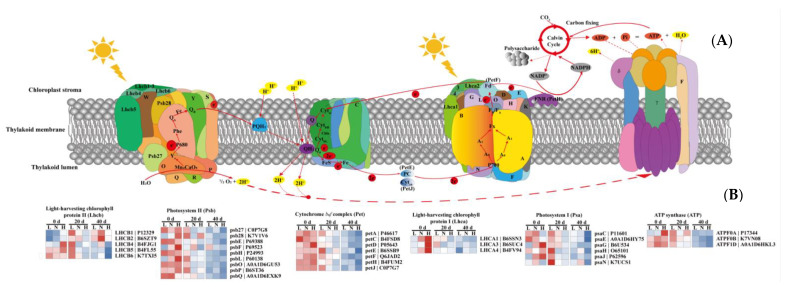
Light-induced changes in the levels of differentially abundant proteins (DAPs) in the photosynthetic apparatus. (**A**) Schematic representation of photosynthetic linear electron flow and proton translocation driven by protein complexes in the thylakoid membrane. (**B**) Levels of differentially abundant proteins (DAPs) in the photosynthetic apparatus. Core light-harvesting chlorophyll protein II and I are Lhcb 1-6 and Lhca 1-4, respectively. Core PSII subunits are Psb27, Psb28, PsbO, PsbP, PsbQ, PsbR, PsbS, PsbW, and PsbY. Core Cyt*b_6_f* subunits are PetC, PetE, PetF, PetH, and PetJ. Core PSI subunits are PsaD, PsaE, PsaF, PsaG, PsaH, PsaK, PsaL, PsaN, and PsaO. Core subunits related to ATP synthase are ATPF, ATP γ, and ATP δ. Colored scales of the *Z* score (–log_10_ *p*-value) are shown: proteins that accumulated at high levels are in red, and proteins with low accumulation levels are in blue. L, low density; N, normal density; H, high density; 0, 20, and 40 d, days after anthesis.

**Table 1 ijms-23-03015-t001:** Mean values ± SD (*n* = 3) of photosynthetic parameters derived from the photosynthetic photon flux density (PPFD) response curves in maize plants grown under low (LD), normal (ND), and high density (HD) planting at 0, 10, 20, 30, and 40 d after anthesis. Different lowercase letters (a, b, c) denote statistical differences by LSD test (*p* ≤ 0.05) between different treatments.

Days After Anthesis (d)	Plant Density (Plants hm^−2^)	A_max_ (μmol m^−2^ s^−1^)	R_d_ (μmol m^−2^ s^−1^)	AQE	LCP (μmol m^−2^ s^−1^)	LSP (μmol m^−2^ s^−1^)
0	LD	44.3 a	6.4 a	0.052 a	123.1 b	975.48 c
	ND	39.6 b	5.9 b	0.043 b	137.9 a	1058.70 a
	HD	38.0 c	4.9 c	0.042 b	117.3 c	1021.17 b
10	LD	38.2 a	5.0 a	0.043 b	117.1 a	1005.0 a
	ND	36.6 b	4.2 b	0.047 a	89.0 b	867.9 b
	HD	35.6 c	3.4 c	0.045 b	74.9 c	866.7 b
20	LD	31.6 a	4.4 a	0.035 b	126.7 a	1029.0 a
	ND	32.8 a	3.8 b	0.040 a	93.9 b	914.6 b
	HD	30.1 b	3.2 c	0.042 a	75.5 c	792.3 c
30	LD	24.0 c	4.0 a	0.030 c	132.5 a	933.8 b
	ND	32.0 a	3.5 b	0.037 b	94.9 b	961.4 a
	HD	27.5 b	2.8 c	0.040 a	70.4 c	757.7 c
40	LD	19.8 c	2.7 a	0.024 c	112.9 a	939.1 a
	ND	23.0 b	2.6 a	0.031 b	84.2 b	824.7 b
	HD	24.2 a	1.7 b	0.035 a	47.7 c	740.1 c

A_max_, maximum gross photosynthetic rate; R_d_, rate of dark respiration; AQE, apparent quantum efficiency; LCP, light compensation point; LSP, light saturation point.

**Table 2 ijms-23-03015-t002:** Definition of terms and formulae for calculation of JIP-test parameters from chlorophyll *a* fluorescence (ChlF) transient OJIP kinetics curves.

Fluorescence Parameter	Description
Measured parameters and basic JIP-test parameters derived from ChlF transient OJIP kinetics curves	
F_0_ = F_20_ _μs_	Minimum fluorescence, when all PSII reaction centers (RCs) are open
F_k_ = F_300_ _μs_	Fluorescence intensity at 300 μs
F_j_ = F_2 ms_	Fluorescence intensity at the J-step (2 ms)
F_i_ = F_30 ms_	Fluorescence intensity at the J-step (30 ms)
F_m_ = F_P_	Maximum fluorescence, when all PSII RCs are closed
V_i_ = (F_30 ms_ − F_0_)/(F_m_ − F_0_)	Relative variable fluorescence at the I-step (30 ms)
V_j_ = (F_2 ms_ − F_0_)/(F_m_ − F_0_)	Relative variable fluorescence at the J-step (2 ms)
W_k_ = (F_300_ _μs_ − F_0_)/(F_j_ − F_0_)	Ratio of variable fluorescence F_k_ to the amplitude F_j_ − F_0_
M_0_ = 4 (F_300_ _μs_ − F_0_)/(F_m_ − F_0_)	Approximated initial slop (in ms^−1^) of the fluorescence transient *V* = *f*(t)
Biophysical parameters derived from the fluorescence parameters	
Specific energy fluxes expressed per active PSII reaction center (RC)	
ABS/RC = M_0_ (1/V_j_) (1/φP_o_)	Absorption flux (of antenna chlorophylls) per RC (also a measure of PSII apparent antenna size)
TR_0_/RC = M_0_ (1/V_j_)	Trapping energy flux leading to Q_A_ reduction per RC (at *t* = 0)
ET_0_/RC = M_0_ (1/V_j_) ψE_o_	Electron transport flux (further than Q_A_^−^) per RC (at *t* = 0)
RE_0_/RC = M_0_ (1/V_j_) (1 − V_i_)	Quantum yield of electron transport from Q_A_^-^ to the PSI end electron acceptors, per RC
DI_0_/RC = (ABS/RC) − (TR_0_/RC)	Dissipated energy flux per RC (at *t* = 0)
RC/CS_0_ = φP_o_ (V_j_/M_0_) (ABS/CS_0_) ≈ φP_o_ (V_j_/M_0_) F_0_	Density of active PSII RCs (Q_A_^−^ reducing PSII RCs) per illuminated cross section (CS) (at *t* = 0)
Quantum yields and efficiencies	
φP_o_ = TR_0_/RC = [1 − (F_0_/F_m_)] = F_v_/F_m_	Maximum quantum yield of primary PSII photochemistry (at *t* = 0)
ψ_o_ = ET_0_/TR_0_ = 1 − V_j_	Probability (at *t* = 0) that a trapped exciton moves an electron into the electron transport chain beyond Q_A_^−^
φE_o_ = ET_0_/ABS = [1 − (F_0_/F_m_)] ψ_o_	Quantum yield for electron transport from Q_A_^−^ to plastoquinone (at *t* = 0)
φD_o_ = F_0_/F_m_ = 1 − φP_o_	Quantum yield (at *t* = 0) of energy dissipation
δR_o_ = RE_0_/ET_0_ = (1 − V_i_)/(1 − V_j_)	Efficiency/probability with which an electron from the intersystem electron carriers is transferred to the reduce end electron acceptors at the PSI acceptor side (RE)
φR_o_ = RE_o_/ABS = [1 − (F_0_/F_m_)] (1 − V_i_)	Quantum yield for reduction of end electron acceptors at the PSI acceptor side (RE)
Performance indexes	
PI_ABS_ = [γ_RC_/(1 − γ_RC_)] [φP_o_/(1 − φP_o_)] [ψ_o_/(1 − ψ_o_)] = (RC/ABS) [φP_o_/(1 − φP_o_)] [ψ_o_/(1 − ψ_o_)]	Performance index (potential) for energy conservation from photons absorbed by PSII to the reduction of intersystem electron acceptors
PI_total_ = PI_ABS_ [δR_o_/(1 − δR_o_)]	Performance index (potential) for energy conservation from photons absorbed by PSII to the reduction of PSI end acceptors

Modified from Strasser et al. [46,87].

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
