# Peer review of "Low Light Increases the Abundance of Light Reaction Proteins: Proteomics Analysis of Maize (Zea mays L.) Grown at High Planting Density"

_ijms, 2022, doi:10.3390/ijms23063015_

Round 1

Reviewer 1 Report

The work described in the manuscript investigates the impact of the density of maize cultures on the photosynthetic capacity of the leafs. Morphological and physiological changes of the leafs have been observed and reported previously and identified to be responsible for changes in the photosynthetic capacity. The main stimulating factor was supposed to be the lower light radiation (LD) under these conditions. In the present work the effect of light intensity due to planting density on light absorption, transmission, and related electron transport was studied by means of fluorescence measurements and proteomics. As a result, densely planted maize showed elevated light utilization through increased electron transport efficiency, promoting coordination between PSII and PSI, as reflected by higher apparent quantum efficiency, lower light compensation point, and lower dark respiration rate. Although I am not an expert of proteomics, the employed methods and the conclusions drawn from the results seem to be sound and may be important for a better understanding of crop growth in the agricultural domain. It has to be stressed that the conclusions remain hypotheses supported by the reported evidence. However, the authors should address in the introduction and the conclusion the influence of other parameters than light intensity. In densely planted maize cultures certainly other factors such as nutrition, water, roots, inter-plant interactions, temperature… might contribute to morphological and physiological modifications/adaptations of the individual organisms. Complementary experiments could involve the artificial variation of light intensity at different plant densities. Since the amount of the present work is already consistent, such experiments could be suggested as a perspective.

Reviewer 2 Report

The authors studied the effect of Low light on the proteomics of maize and found that the abundance of light reaction proteins increases under this situation. The work is well performed and presented, but some issues need to be concerned before acceptance.

1. More details need to be added in the figure legends, including analysis method for significance, sample sizes, and replication numbers.

2. The authors used the proteins with 1.3 and 0.77 change folds for analysis, please include the explanation in the manuscript. 

3. It is interesting to detect the overlap of the different expressed proteins among three sample sets. Venn diagrams could be added to show the overlap among samples and among sample sets.

4. The authors claimed the major changes on the components of photosynthesis system, but the changes varied among samples, it would be better to measure additional photosynthesis parameters to confirm these conclusions.  
